# Consumer Attitude towards Sustainability of Fast Fashion Products in the UK

**Bo Zhang [1], Yaozhong Zhang [2,3] and Peng Zhou [4,\*]**

1   School of Economics and Management, Beijing University of Chemical Technology, Beijing 100029, China;
    bzhang@mail.buct.edu.cn
2   School of Art and Design, Shenyang Normal University, Shenyang 110034, China;
    st20163354@outlook.cardiffmet.ac.uk
3   Cardiff School of Management, Cardiff Metropolitan University, Cardiff CF10 3EU, UK
4   Cardiff Business School, Cardiff University, Cardiff CF10 3EU, UK
\*   Correspondence: zhoup1@cardiff.ac.uk; Tel.: +44-(0)2920-688778

**Abstract:** This paper attempts to provide an up-to-date depiction and analysis of the consumer's attitude towards sustainability of fast fashion products in the UK. Four related strands of literature are reviewed to establish a tri-component model of attitude (ABC), i.e., Affective, Behavioural and Cognitive. A wide set of determinants for attitude is identified, including income, price, gender, culture, religion, age, etc. Based on this conceptual framework, an online questionnaire is designed and sent to university students and alumni in the UK, returning 128 valid responses. Both descriptive statistics and regression analysis (oprobit) are employed to shed light on the three components of attitude towards sustainability. It is found that cognitive and behavioural components converge across cultures and religions, but the affective component remains significantly diverse. Employment status contributes to the awareness, decision and feeling of sustainability features, but gender only matters for purchase decisions. In general, there is an improved cognitive and affective awareness of sustainability, but this does not automatically translate to purchase behaviour. Policy interventions like taxes and subsidies are still needed to foster sustainability in the fast fashion industry.

**Keywords:** sustainability awareness; fast fashion; consumer attitude

## 1. Introduction

The fashion industry is reportedly the world's third biggest manufacturing industry behind automotive and technology industries [1]. Over 150 billion garments are produced in the world each year [2]. According to a recent report by the House of Commons [3], people buy more clothes per person in the UK (26.7 kg) than any other country in Europe, e.g., Germany (16.7 kg), Denmark (16.0 kg), France (9.0 kg), Italy (14.5 kg) and the Netherlands (14.0 kg) according to ECAP [4]. As a result, the fashion industry in the UK grows at a faster rate than the rest of the economy (5.4% versus 1.6% in 2016), accounting for £32 billion in 2017. The fashion industry is also a large employer of the British labour force—about 890,000 people work in retail, manufacturing, brands, and design businesses related to fashion products [5]. All these facts make the UK a very interesting case for studying attitude towards sustainability in the fashion industry.

Among others, the "fast fashion" business model is a salient success in this trend, because its low prices and fast product rotations encourage over-consumption. By the mid-1970s, many fashion brands began rapidly copying catwalk styles, producing garments at a much lower costs and supplying cheap fashion products on retailing markets within months [6,7]. This business model gained popularity throughout the 1980s, and some described it as the "democratisation of fashion", because the once-exclusive luxuries were now accessible to everyone. In the 1990s the fast fashion industry became mature and many leading brands such as Zara, H&M and GAP have established their positions in

fashion markets all over the world [8]. The rise of fast fashion gained another surge in 2005, when the World Trade Organisation eliminated the quota system of outsourcing abroad. By making use of cheap labour and materials all over the world (especially in China and India), fashion become a huge, globalised business. This is true not only for fast fashion brands but also for luxury fashion; where there were once only two collections per year, there are now six or more.

However, at the same time, the fashion industry (especially fast fashion brands) also generates huge waste and pressures on the environment [9]. More than \$500 billion is lost worldwide every year due to clothing underutilisation and the lack of recycling [10]. It is projected that by 2030 global apparel consumption will rise from 62 billion tons today to 102 million tons [11]. Moreover, wastes generated by fashion products, such as textiles, chemicals, and dyes, impose environmental damages and climate change pressures. In the current unidirectional globalised supply chain, fashion products' carbon footprint is one of the largest, creating even more greenhouse gases than aviation and shipping industries combined because almost all fashion products are outsourced and transported internationally. It is estimated that, if the full lifecycle of clothing is considered, the fashion industry is responsible for 3.3 billion tonnes or 10 per cent of global $CO_2$ emissions and 20 per cent of global waste streams [12].

This worrying trend places sustainability in the spotlight of policy discussions and research agendas. To commit and contribute to sustainability, the UK signed onto the United Nations Sustainable Development Goals in September 2015. Three environmental initiatives have been developed in the UK since then: waste disposal regulations, consumer education and post-consumer recycling programmes. To meet the future carbon budgets and reach net zero emission by 2050, the UK will have to change its consumption patterns by consumers and improve the resource productivity of producers [3]. In academic literature, increasing attention has been paid to the environmental impacts of fast fashion products and the implications for consumer perception of product quality [13,14]. It is observed that fast fashion business model usually leads to low quality products [15]. A recent Unilever study [16] found that over one third of consumers (33% of 20,000 adults in five EU countries, including the UK) are now choosing to buy from brands they believe are doing social or environmental good. 53% of shoppers say they feel better when they buy products that are sustainably produced. 21% of the people surveyed would actively choose brands if they made their sustainability credentials clearer on their packaging and in their marketing. Unilever estimates that this represents a potential untapped opportunity of €966 billion out of a €2.5 trillion total market for sustainable goods [17].

Logically, there are three perspectives to look at the relationship between fast fashion and sustainability: the supply side (fast fashion brands), the demand side (consumers) and the regulator side (governments and industrial organisations). This research focuses on the consumers' perspective and aims to sketch a clear and up-to-date picture of how consumers make decisions under different degrees of awareness of sustainability. Therefore, the paper has two main contributions to the literature: one theoretical and one empirical. On the one hand, three related strands of literature are reviewed to establish a unified conceptual framework for analysing the components and factors of attitudes towards sustainability. On the other hand, a structured questionnaire is carefully designed to collect a representative sample and to empirically investigate the status quo of the attitudes towards sustainability in the fast fashion industry in the UK. The key novelty of our research is the application of an ordered probit econometric model to quantify the effects of different factors on the three components of attitude towards sustainability. This is a first in marketing literature to our knowledge.

The structure of our paper is as follows. The research context in the Introduction section identifies the UK as a good case study for studying the customer attitude towards sustainability in the fast fashion industry. In the Literature Review section that follows, we review the three strands of literature to summarise different components and factors of attitude. Section 3 operationalises the conceptual framework into a data strategy to address

the stated research aim and objectives. A carefully designed online questionnaire quantifies measures of the components and factors of attitude towards sustainability. Section 4 therefore analyses the data collected and draws discussions and conclusions.

## 2. Literature Review

### 2.1. Literature on Fast Fashion

Fast fashion is usually treated as an accelerated business model featured with short product lifecycles, catwalk fashion imitation (trickled-down trends) and affordable prices [7]. The most successful brands in this segment include Spanish conglomerate Zara and Swedish counterpart H&M. They can translate a fashion idea to a fashion product within two–three weeks, which results in up to 24 collections a year [6], in contrast to the high-end luxury fashion brands with only one–two collections a year [18]. In the first decade of the 21st century, the concept of fast fashion revolutionised the fashion industry, including some luxury brands, in the frequency of collection release [19]. Online shopping services have especially helped young, middle-class female consumers to fulfil their demand for new fashion styles [20]. Retailers such as Zara, H&M and Topshop are known for designing fashion products to be used less than 10 times and encouraging so-called "throwaway fashion" [8], which is criticised by ethical consumers [21].

A key element of the fast fashion business model is its globalised supply chain. To respond in a timely fashion to emerging market trends and needs, fast fashion manufacturers adopt a quick response strategy and supply chain network, which enable a prompt information flow and accurate forecast of the market. In this system, brands like Zara and H&M can arrange sourcing and logistics as close to the release date as possible. Orders are repeatedly placed and updated throughout seasons of a year. This is very different from the pre-season ordering system in traditional retailers [22].

As a response to the criticism of the waste and pollution resulting from fast fashion, the concept of "slow fashion" is developed to help consumers to consider sustainable practices related to fashion production, distribution and use [23]. It encourages consumers to "value and know the object" [24] and integrates experience with self-enhancement values [25]. As opposed to fast fashion, which generates large volumes of waste and environmental pollution, the slow fashion model pays special attention to sustainability in design, production, consumption and use [26].

It is well recognised that fast fashion companies have recently put more effort into environment-friendly collections and branding, focusing on sustainability. For example, H&M launched "the Conscious Collection" created from sustainable material [27] and Zara designed its first sustainable product line in 2016, the "Join Life" materials [28]. Moreover, both companies and most fast fashion retailers provide rich information about their work with sustainability on their websites. Sustainable fashion, which is similar to the idea of slow fashion, is a new trend within the fast fashion industry [29]. Nevertheless, the industry's low prices stimulate increased consumption and thereby have a higher environmental and social impact [30].

In addition to the indispensable role of fashion companies, recent literature shows that consumer-related factors, such as lack of consumer awareness, inappropriate retail environment and social norms are crucial to moving away from fast fashion to sustainable fashion [24]. Consumer's adoption of an eco-conscious fashion acquisition depends on consumer awareness through education on reducing waste and environmental impact [31]. Thus, it is important to understand how consumer's ethical values are shaped by different factors so that informative guidelines for sustainability in fashion products can be provided [32]. Therefore, we intend to explore consumers' attitudes towards sustainability in fashion product purchases, which entails a review of the concept of sustainability and the research on sustainability in fashion studies.

## 2.2. Literature on Sustainability

Sustainability, as a primary issue of the 21st century, has many definitions. The term sustainability was coined in 1987 in Brundtland report, and its original meaning is "satisfying the current needs without compromising the future generation's needs" [33]. Since then, sustainability is extended to encompass three perspectives: environmental, economic and social, known as the "Triple Bottom Line" of sustainability [34]. In current literature, sustainability refers to activities that can be continued indefinitely without causing harm to the environment, the way that you expect to be treated and you treat others, and the consideration that meets a current generation's needs without compromising those of future generations [35,36]. As summarised by Seidman [37], sustainability is about much more than our relationship with the environment; it's about our relationship with ourselves, our communities and our institutions. Indeed, the word "environment" is defined with human beings at the implicit centre, so sustainability is essentially about trade-offs between different groups of people, such as the rich and the poor, the young and the old, labourer and capitalists and developed and developing countries. Sustainability involves complicated and dynamic interactions between human livelihood and the environment. It permeates through ecological, economic, social and political dimensions, locally, regionally and globally [38]. Therefore, Joy et al. [13] describe sustainability as a "social contract" between a business with the society.

Production processes of textile and garments in fashion industry impose many concerns on sustainability [39]. For example, there is a large amount of energy use and water consumption, greenhouse gas emission, hazardous waste generation, and discharge of toxic effluent containing dyes, finishes and auxiliaries to the ecosystem [40,41].

Combining the concept of sustainability with the fashion industry, sustainable fashion is defined as fashion products with a conscience to care about labour conditions and environmental responsibility [14]. An increasing proportion of consumers advocate purchasing sustainable fashion products as a way to meet their psychological needs such as the attitudes of equality and sustainability [42]. There are four aspects of sustainable fashion in existing literature: (i) sustainable production and remanufacturing [43,44], (ii) green marketing [45,46], (iii) green information sharing [47], and (iv) green attitude and education [48]. The first three are responsibilities of fashion companies, while the last one is about fashion consumers.

In business management literature, sustainability is usually paired with corporate social responsibility [49,50], which is important to a company's strategy [51,52]. It is widely recognised that corporate social responsibility has significant effects on a firm's competitive advantage over its opponents and market shares [53]. Therefore, even if many aspects of sustainability are related to companies, it is inextricably intertwined with consumers' preferences and values. Hur and Cassidy [54] verify that there are both internal (from the fast fashion designers) and external (from the fast fashion customers) challenges to incorporating sustainability into the fashion design process. Therefore, it is important for fashion companies to understand the trend of consumer attitude toward sustainable fashion to compete in the market. To gain a deeper insight into how sustainability enters consumer's decision-making in fashion products, we will extensively review consumer behaviour theory in economics, psychology and management.

## 2.3. Literature on Consumer Behaviour

Consumer behaviour theory explores how consumers make decisions. Sustainability is an attribute of products explicitly or implicitly relevant to consumer buying decisions. Therefore, it is vital for businesses to understand the mechanism of the process and factors that affect the process to design and deliver their products.

### 2.3.1. Economic Theory of Consumer Behaviour

Fundamentally, fast fashion products are still economic goods. Therefore, the demand for fast fashion products follows basic economic laws—a higher price leads to lower

demand and a higher income leads to higher demand. In our questionnaire, these economic factors will be collected in Section 3 and fed into the empirical model in Section 4. A common assumption of almost all modern economic interpretations is that consumers are rational optimisers. They know exactly what they want ("utility function") and what they have ("budget constraint"), which are usually described in a mathematical model as shown in panel (a) of Figure 1. The diagram shows that, as price drops for sustainable fashion products, the budget constraint (the affordability of the consumer) is relaxed and it shifts outward, resulting in a higher utility level (from point A to B to D). Each optimal demand corresponds to a point along the demand curve, which contains all the possible combinations between price and optimal quantity of demand. On the supply side, there is a similar optimisation decision for the producers resulting in a supply curve. The market interactions between consumers and producers determine the general equilibrium price and quantity (panel (b) of Figure 1). In short, the wisdom of economics shed light on the most important two factors underlying consumer's decision-making, i.e., price and income.

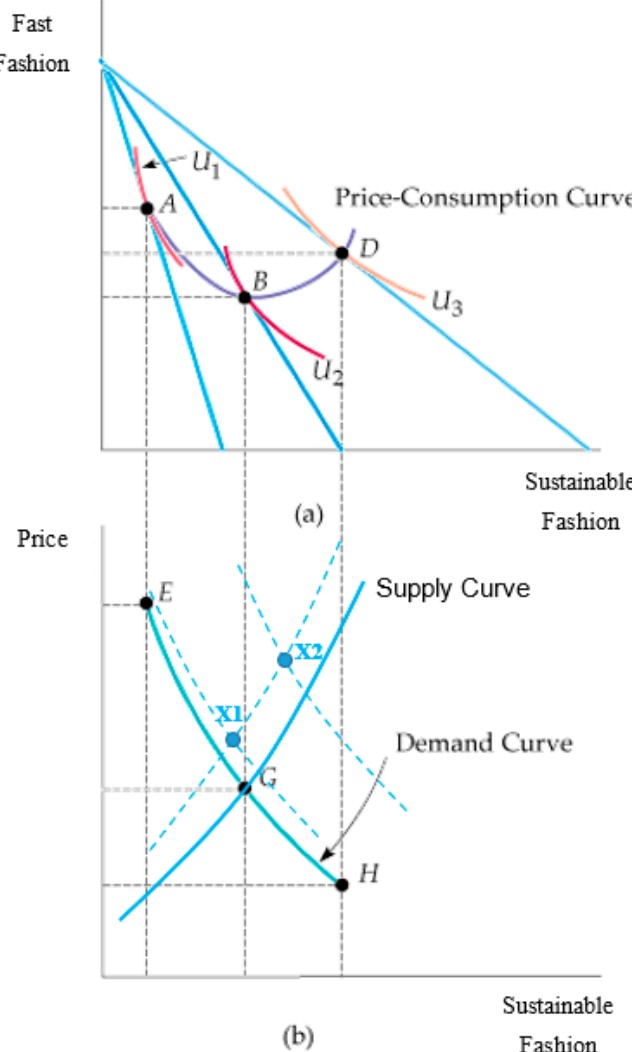

**Figure 1.** Consumer Optimisation Behaviour and Market Equilibrium. Panel (**a**) is the price consumption curve and Panel (**b**) is the supply-demand curves.

In our context, sustainability obviously affects both subjective preferences and objective prices. On the one hand, as more people are aware of environmental issues and importance of sustainability, consumers tend to prefer natural, durable materials (sustainable fashion) rather than artificial, non-biodegradable materials (fast fashion) in clothes. The

demand curve for the eco-friendly clothes shifts out. On the other hand, most eco-friendly materials are more expensive, so a higher price may deter customers from purchasing clothes with sustainability features. As a result, the supply curve for the eco-friendly clothes shifts up. These two forces are opposite and cancel out each other. Two possibilities of the final effect are shown in X1 and X2 of Figure 1. The net effect on the total demand (originally at G) depends on which side dominates.

2.3.2. Psychological Theory of Consumer Behaviour

Traditional economic interpretations of consumer behaviour, however, ignore the emotional aspect of buying activities. It is effectively a normative analysis (what people should do) rather than a positive analysis (what people actually do). Humans are not machines and we do not always make rational choices as economics predicts.

To complement the omissions from economics, psychology offers a different perspective to understand consumer behaviour, including cognitive, emotional, and social needs. As a precursor, the sociocultural theory emphasises the roles of social interactions and language influences [55–57] since all knowledge is socially constructed and perceived. Furthermore, the psychoanalytic theory developed by Sigmund Freud stresses the struggle among id, ego and superego to meet personal and social needs. Inspired by Freud, Bernays [58] explores the irrational forces underlying consumption behaviour and he successfully applied his theory to a smoking campaign for female consumers. As Freud's student, Carl Jung develops the concept of neurosis, which is referred to as a significant unresolved tension between contending attitudes [59]. A tenet of Jung's approach is that individual context shapes meaning, and from meaning comes behaviour. This notion is especially pertinent to fashion products, which also have "value in possession" [60]. For example, emotions are the core of luxury, from simple pleasure to self-identity to social comparison (the so-called "conspicuous consumption" by Veblen). Sustainability and corporate responsibility are obviously important elements of consumer's perception of values in possession of fashion products.

In contrast to economics, psychology defines values from the internal, subjective point of view rather than from the external, objective perspective. Following the same vein, Maslow proposes an influential hierarchical analysis of human needs. Different levels of consumption products belong to different levels of needs. For example, food and clothing are physiological needs, but gym and fashion are belongingness-and-love needs. Consumers who do not resolve the lower needs get stuck in that level. Sustainability can be treated as part of belongingness, love and esteem needs, because ethical and responsible consumption is beyond individuals.

2.3.3. Anthropological Theory of Consumer Behaviour

In contrast to economics and psychology which focus on short-run patterns, anthropology gives a long-run explanation of the trend in consumer behaviour through the lens of culture. It is argued that the slow evolution of every culture reflects the process of status understanding [61]. Building on this understanding, Ruth Benedict [62] regards consumerism as an unnatural culture and the learnt consumer behaviour is projected symbolically through brands and marketing communication to push us to consume. To a large extent, culture determines how members of society think and feel, so it directs actions and defines shared beliefs. Therefore, culture is essentially a system of communication in the society [63]. In marketing practice, storytelling is used to create and communicate brand image to the consumers.

Sustainability was not a heatedly discussed issue until long-run issues facing the human civilisation became salient in the 20th century. Customer preferences evolve over time to account for this new culture of environmentalism at a slow but steady pace. This trend will only be reinforced in the future given the scarcity of natural resources.

### 2.3.4. Marketing Theory of Consumer Behaviour

The wisdom reviewed in the three disciplines (economics, psychology and anthropology) provide three complementary perspectives to understand consumer behaviour for modern marketing. As a summary, we use Figure 2 to model the decision-making process of a consumer and the influencing factors.

If we treat buying decisions as essentially an economic decision, then perceived values of the product purchased are basically given. The only variables to consider are prices of the products and its alternatives as well as the income of consumers. However, if we use psychological and anthropological perspectives to analyse the formation and evolution of perceived values [64], we will be able to identify a wider range of factors to influence the consumer's decision, such as personal, social, cultural, psychological and situational factors [65]. One particularly important factor in modern consumption is information search. Most people are significantly influenced by search engine and social media when they shop [66]. That is why commercial campaigns include digital marketing tools as an indispensable part. Once purchase decisions are made, consumers will rank products in their evoked set according to pre-set criteria (or "preferences orderings" in economic jargon) and select the most desirable product (or "optimisation" in economic jargon). Then, actual purchases occur, but marketers still try to influence the consumers with offers right up until the purchase. However, this is not the end, as consumers will automatically provoke post-purchase evaluation of the product and feed forward to future purchases.

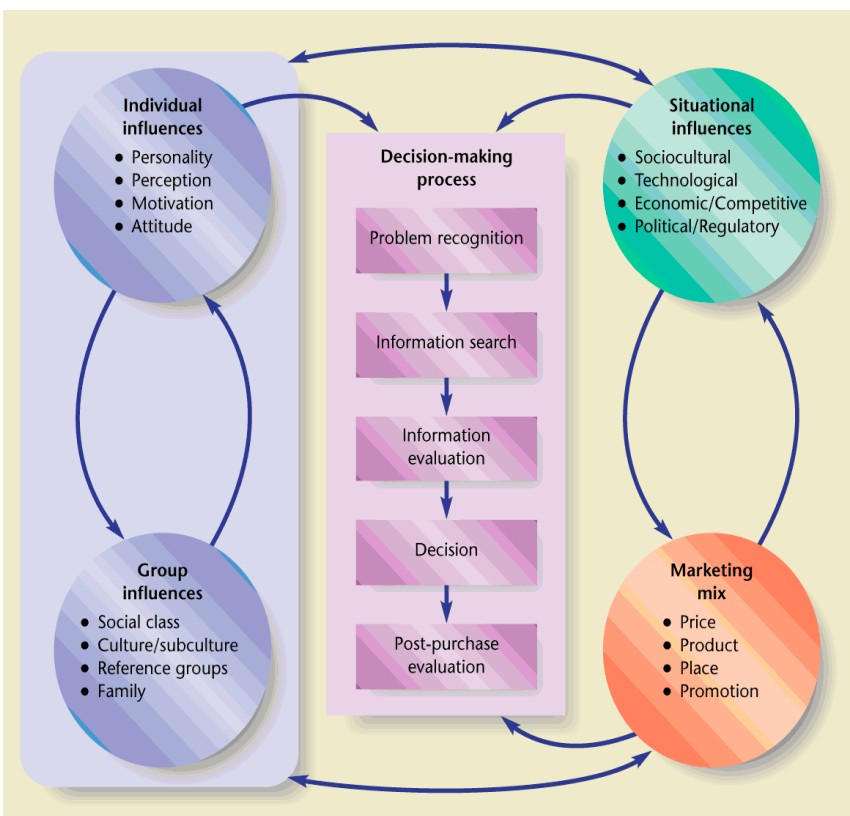

**Figure 2.** The Consumer's Decision-Making Process and Influencing Factors. Source: Blackwell et al. [67].

In the process of consumption decision-making, different individual units can be involved. A decision-making unit (DMU) can be an initiator (who begins the process of considering a purchase), an influencer (who attempts to persuade others in a purchase), a decider (who has the power to make the decision), a buyer (who conducts the transaction), a user (who actually use the product), a financier (who pays) and a gatekeeper (who

discloses information). This detailed refinement of a "consumer" of different roles provides a useful insight into different buying situations.

There are different types of consumer buying situations. In terms of the degree of reflection in the decision making, the most common type is routine rebuy, including items that are bought frequently on a regular basis. For example, toilet rolls are usually purchased without much thought involved. In this type of purchase, needs and prices are the two primary factors to consider, and the purchase has little to do with cultural or social factors. However, during the COVID-19 pandemic, toilet rolls became in shortage due to psychological sentiment and social herding behaviour. In this case, it is no longer a routine rebuy, but a panic hoarding. The second type of buying is modified rebuy, in which case consumers are familiar with a range of alternatives and may choose different brands in purchases. Fashion products are a good example of this type, since consumers may want to change brands from time to time to reflect their versatility in tastes. The third type is completely novel purchases, i.e., there is no previous experience of the product. This is also a common phenomenon in sustainable fashion products, given that consumers may have an open mind in new fashion trends and needs.

Another popular criterion to categorise buying situations is the level of involvement of consumers in the purchase. Factors that affect the level of involvement include self-image (how we see ourselves), perceived risks (e.g., financial, physical, functional, social, and psychological), associated costs of purchase (e.g., return policy, [68,69]), social factors (e.g., wealth, religion, habits, education, family size) and hedonism (pleasure seeking). For example, routine rebuy has low involvement because it is habitual and there are few differences between brands, while complex buying has high involvement because it is information intensive and there are significant differences between brands. Sustainable fashion buying belongs to the latter category.

Specifically, our study concentrates on the individual influences of consumer behaviour (upper left corner of Figure 2). Together with motivation and perception, attitude offers a general, lasting evaluation of any person, object, advertisement or product. If we want to explore whether and how much attitude towards sustainability affects fast fashion purchases, it is necessary to elaborate on the conceptual framework of attitude to shed light on our empirical exercise.

An attitude is a relatively enduring organisation of beliefs, feelings and behavioural tendencies towards socially significant objects, groups, events or symbols [70]. It is a psychological tendency that is expressed by evaluating a particular entity with some degree of favour or disfavour [71]. Attitudes structure can be described in terms of three components.

- Affective component: this involves a person's feelings or emotions about the object. For example: "I don't like fast fashion products".
- Behavioural component: the way the attitude influences on how we act or behave on an object. For example: "I will not buy any fast fashion products".
- Cognitive component: this involves a person's belief or knowledge about an object. For example: "I believe fast fashion products are not sustainable".

This model is known as the tri-component or ABC model of attitudes. In the next section, this model of attitude will be utilised to develop the three research objectives to underpin the research aim as outlined in the next subsection.

## 3. Methods

### 3.1. Research Objectives/Questions

The three strands of literature reviewed have established two solid conceptual frameworks for analysing sustainability in the fast fashion industry. One the one hand, the ABC theory of attitude lays out a conceptual framework for the *components* of attitude. This is to define "*what is the attitude*" towards sustainability. On the other hand, a conceptual framework for the *factors* of attitude is also developed, including economic (e.g., price, income), psychological (e.g., different hierarchies of needs), anthropological (e.g., religion, culture)

and marketing (e.g., perceptions) factors. This is to define "*what affects the attitude*" towards sustainability. However, these conceptual frameworks are all theoretical. The empirical investigation of the consumer attitude towards sustainability is still scanty [72], especially for the UK which is one of the biggest fast fashion markets in the world. This real-world information is important for consumers to make purchase decisions, for producers to make business plans, and for the policymakers to design sustainability policies in the fast fashion industry in the UK.

Based on the literature review of fast fashion, sustainability and consumer behaviour, we determine the following research aim of this paper to fill the gap in the literature:

- Research Aim: to explore consumer attitude towards sustainability of fast fashion products in the UK.

To achieve this research aim, three research objectives (or sub research questions) are established in accordance with the tri-component or ABC model of attitude summarised in the last section. The three research objectives (or research questions) correspond to affective, behavioural and cognitive components of attitude respectively:

- Research Objective 1 (Cognitive): To investigate how much customers are aware of sustainability in the fast fashion industry. The equivalent research question is "*To what extent customers are aware of sustainability in the fast fashion industry in the UK?*"
- Research Objective 2 (Behavioural): To explore how consumers make decisions on fast fashion products with sustainability features. The equivalent research question is "*What are the factors that affect consumers' purchase decisions on fast fashion products with sustainability features?*"
- Research Objective 3 (Affective): To understand how customers feel about fast fashion products with sustainability features. The equivalent research question is "*How do customers feel about fast fashion products with sustainable features?*"

A structured questionnaire will be carefully designed according to the ABC conceptual framework to shed empirical light on these three research objectives/questions.

### 3.2. Research Design

Taking a positivist ontology in research philosophy, it is believed that the attitude of British consumers can be observed and measured to a satisfactory degree [73]. However, we are aware that attitudes are subjective and subject to constant changes, so the conclusions based on the empirical observations are not eternal or generalisable. Therefore, the ultimate purpose of the study is to describe a corner of the big picture for a specific point in time, rather than striving for a universal truth. It implies that the attitude of British consumers on fast fashion is socially constructed through culture and language, so our epistemological stance is a positivist viewpoint bended towards interpretivism. Thus, the axiology of this research lies in the middle between objectivism and subjectivism, since the value of sustainability is embedded in the conception and design of the research even if the questions in the questionnaire are worded as value-free as possible.

Given the confirmatory nature of the study, it is appropriate to undertake a mono quantitative method. This is done by collecting quantitative data from an online questionnaire. A random sample of young consumers of fast fashion products based in the UK are asked a set of questions relating to each of the main research objectives (e.g., feeling, choice and knowledge about fast fashion products with sustainability features) as well as some individual attributes (e.g., gender, age, occupation, location, etc.). The questions are designed to reflect the conceptual framework reviewed in economic (e.g., price and budget), psychological (e.g., different needs), anthropological (e.g., cultural and religious background) and marketing theories (e.g., attitude) in the literature section.

The questionnaire (attached in Appendix A) is created using Google Form, and publicised to university students and alumni via social media platforms (e.g., Facebook, Twitter, Instagram and Wechat) of Student Unions in the UK. Making use of the international student community in the UK, the data collected can well represent the young generation

(18~27 years old or the "Gen Z") in terms of gender, nationality, religious background and employment background. It is arguable that this age group is the main customers of fast fashion products and their behaviour and attitude are very similar across the world thanks to the Internet and social media [13].

After a five-day data collection duration, we received 128 valid responses. Due to the anonymity and intractability of online users, we are not able to determine the response rate exactly, but we can infer the number of reads from the analytics of social media platforms. The total number of reads across all the social media platforms is 8734, so the response rate is $\frac{128}{734} = 17.4\%$. This is an underestimation of the actual response rate because many online users are not really paying adequate attention to the posts as in the traditional questionnaire collection.

The collected data are then analysed using two statistical techniques following an inductive reasoning approach. The first is descriptive analysis to summarise the up-to-date situation of the British customer's attitude towards sustainability, while the second technique is regression analysis to establish some causal relationship between different factors (individual attributes) and the three components of attitude.

Measurement is a tricky issue given that the attitude is quite subjective. Following the convention in literature [73], we adopt a five-scale measure, i.e., strongly disagree (1), disagree (2), neutral (3), agree (4) and strongly agree (5), to quantify attitude. The questionnaire is attached in the Appendix A.

### 3.3. Ethics

Given that this project involves primary data collection, it is essential to consider the ethical issues involved before the questionnaire is sent out. To ensure the appropriateness of the process, ethical approval is sought from the research committee of Cardiff Metropolitan University. The participation in the questionnaire is voluntary and withdrawal is permitted at any stage of the data collection. Separate consent forms are designed and signed by the participants before taking part of the survey. In addition, there is a preamble disclaimer section at the beginning of the questionnaire to ensure the participants are aware of the purpose of the project, the voluntariness of the questionnaire, the prerequisites for the participants and the contact.

### 3.4. The Data

Among these 128 respondents, we have 44.53% males and 55.47% females (the UK population gender ratio is 49.2% males and 50.8% female, [74]). Age ranges from 18 to 27 years old with an average of 22.75. Most of the respondents are British (44.53%) and Asian (15.63%). Religion-wise, Christian (57.81%) and no religion (23.44%) are the most popular answers. The distribution of religion is quite close to that of the UK population [74]. As predicted, most of our sample are students (77.34%) with some of the graduates working either full-time (4.69%) or part-time (8.59%). A budget question is also asked to categorise the respondents by their economic background. It is interesting to see a difference in expenditure pattern between male and female customers. As tabulated in Table 1, the males have a bimodal distribution of budget share for fast fashion products. Most of them reserve 10–20% of their budget for fast fashion products (21.88%), while the second most popular choice is above 30% (18.75%). The lowest and the middle categories (under 10% and 20–30%) are much lower (5.47% and 9.38% respectively). In contrast, female customers have a more uniform distribution, which peaks at 10–20% but remains relatively stable.

**Table 1.** Cross Tabulation of Budget Share of Fast Fashion by Gender.

| Budget Share | Male | Female | Total |
|---|---|---|---|
| Under 10% | 5.47% | 5.47% | 10.94% |
| 10–20% | 21.88% | 14.84% | 36.72% |
| 20–30% | 9.38% | 11.72% | 21.09% |
| Above 30% | 18.75% | 12.50% | 31.25% |
| | 55.47% | 44.53% | 100% |

## 4. Results

As discussed in last section, we are going to employ both descriptive statistics and regression analysis to achieve the research aim, i.e., to explore consumer attitude towards sustainability of fast fashion products in the UK. Findings are organised by the three research objectives.

### 4.1. Cognitive

The first substantial section of the questionnaire focuses on investigating the awareness and knowledge about sustainability in fast fashion industry (Research Objective 1). It is the cognitive component of the ABC model of attitude.

Three issues related to the cognitive component of attitude towards sustainability in the fast fashion industry are asked:

- Q7: I am aware of social equity issues in the fast fashion industry such as working conditions of factory worker and fair trade.
- Q8: I am aware of child labour and sweatshop issues in the global supply chain of the fast fashion industry.
- Q9: I am informed of environmental issues in the fast fashion industry such as waste and pollution caused by excessive production of garments.

We tabulate the distributions of cognitive knowledge in Table 2. It is found that females tend to pay more attention to child labour issues (Q8) and environmental issues (Q9), while males are more aware of social equality issues (Q7).

**Table 2.** Cross Tabulation of Cognitive Questions by Gender.

| | Q7 | | | Q8 | | | Q9 | | |
|---|---|---|---|---|---|---|---|---|---|
| Scale | Male | Female | All | Male | Female | All | Male | Female | All |
| 1 | 0.00% | 0.00% | 0.00% | 0.78% | 0.00% | 0.78% | 0.00% | 0.00% | 0.00% |
| 2 | 3.91% | 0.00% | 3.91% | 4.69% | 4.69% | 9.38% | 2.34% | 2.34% | 4.69% |
| 3 | 13.28% | 11.72% | 25.00% | 11.72% | 11.72% | 23.44% | 7.81% | 9.38% | 17.19% |
| 4 | 17.97% | 14.84% | 32.81% | 20.31% | 14.84% | 35.16% | 21.09% | 10.94% | 32.03% |
| 5 | 20.31% | 17.97% | 38.28% | 17.97% | 13.28% | 31.25% | 24.22% | 21.88% | 46.09% |
| Mean | 3.99 | 4.14 | 4.05 | 3.90 | 3.82 | 3.87 | 4.21 | 4.18 | 4.20 |

As shown in the correlation coefficient matrix (Table 2), those who are more aware of one issue is likely to be aware of other issues, because the correlation coefficients among Q7, Q8 and Q9 are all positive.

In addition, another two questions are asked to retrieve the level of knowledge of the respondents:

- Q10: I am knowledgeable about the apparel brands that sell eco-friendly fashion products.
- Q11: To your knowledge, which of the following fabrics is the most eco-friendly? 1: Cotton; 2: Synthetics; 3: Wool; 4: Modal.

Note that Q10 is a question on the *claimed* level of knowledge, while Q11 is a question to identify the *actual* level of knowledge in sustainability. The correct answer is "4. Modal",

so anyone who claims that s/he is knowledgeable in Q10 is expected to answer Q11 correctly. It is true according to Table 3, where Q10 and Q11 have a positive and significant value of 0.342. Also, the correlation coefficients between the actual knowledge (Q11) and the awareness of sustainability issues are stronger than those between the claimed knowledge (Q10) and the awareness.

**Table 3.** Correlation Coefficient Matrix for Cognitive Questions.

|  | **Q7** | **Q8** | **Q9** | **Q10** | **Q11** |
|---|---|---|---|---|---|
| Q7 | 1 |  |  |  |  |
| Q8 | 0.0351 * | 1 |  |  |  |
| Q9 | 0.0037 * | 0.0923 ** | 1 |  |  |
| Q10 | 0.0369 * | 0.0325 * | 0.1459 * | 1 |  |
| Q11 | 0.1343 *** | 0.0632 ** | 0.2801 *** | 0.342 *** | 1 |

Significance levels: * 10%, ** 5%, *** 1%.

Findings above are all based on descriptive statistics of one variable or a pair of variables. To fully capture the pure causal relationship between individual characteristics and their awareness of sustainability, we need to use regression models to control for different factors at the same time.

Given that the dependent variables (Q7, Q8, Q9) are ordinal and categorical, we adopt the Ordinal Probit (oprobit) model to capture the determinants for awareness of sustainability among the young generation (Table 4).

**Table 4.** Oprobit Regression Results for Cognitive Questions.

|  | **Regressors** | **Q7** | **Q8** | **Q9** |
|---|---|---|---|---|
| Q1 | Female | −0.259 ** | 0.041 | −0.080 |
| Q2 | Age | 0.012 | −0.021 | −0.077 * |
| Q3 | European | 0.172 * | 0.000 | 0.240 ** |
|  | American | 0.130 | 0.378 * | 0.436 ** |
|  | Oceanian | −0.242 | −0.108 | 0.238 |
|  | African | 0.545 ** | −0.066 | −0.077 |
|  | Asian | 0.420 | −0.307 * | 0.370 |
| Q4 | Islam | 0.156 | −0.259 | −0.273 |
|  | Buddhism | 0.891 * | 0.566 * | 0.880 |
|  | Hindu | 0.147 | 0.084 | 0.565 * |
|  | Sikhism | 0.265 | 0.554 * | −0.232 |
|  | No Religion | −0.567 * | −0.699 * | −0.182 |
| Q5 | Part-time worker | 0.665 * | 0.342 | 0.998 ** |
|  | Self-employed | 4.355 ** | −0.098 | 3.663 ** |
|  | Unemployed | −0.450 ** | −0.873 ** | −1.789 ** |
|  | Student | −0.650 * | −0.338 * | −1.375 * |
|  | Other | 0.236 | 0.227 | −0.223 |
| Q6 | Budget Share | −0.046 * | −0.079 * | −0.112 ** |
| Q10 | Claimed Knowledge | 0.038 * | 0.073 * | 0.133 ** |
| Q11 | Actual Knowledge | 0.110 ** | 0.034 * | 0.160 ** |
|  | No. of Obs. | 128 | 128 | 128 |
|  | AIC | 334.359 | 372.84 | 325.369 |
|  | BIC | 397.104 | 438.436 | 388.114 |

Significance levels: * 10%, ** 5%.

As a summary, we list the key findings on the cognitive component of the attitude towards sustainability (Research Objective 1) in the following.

- Females pay more attention to child labour issues and environmental issues, while males are more aware of social equality issues.

- Those who work tend to be more aware of the sustainability issues than those who do not.
- Awareness drops as the budget share of fast fashion products rises.
- Claimed and actual knowledge on sustainability do not always equate, and it is the actual knowledge that contributes to the awareness.
- Factors such as age, nationality and religion are not significant to the cognitive component of attitude towards sustainability.

*4.2. Behavioural*

As reviewed in the literature, economic decisions depend on preferences and income. Other factors like information, social environment and feelings are introduced by psychological and marketing literature. In this subsection, we descriptively summarise the consumers' decisions of fast fashion products, and then use regression analysis to identify the factors driving the decisions (Research Objective 2).

Three issues related to sustainability in the fast fashion consumption are asked:

- Q12: From all your clothing purchases, what percentage is sustainable? (1) Under 10%; (2) 10–20%; (3) 20–30%; (4) 30–40%; (5) 40–50%; (6) Above 50%.
- Q13: How much percent more would you like to pay for a fast fashion product with sustainability features? (1) Under 1%; (2) 1–5%; (3) 5–10%; (4) 10–20%; (5) 20–50%; (6) Above 50%.
- Q14: How much percent more income is needed before you can consider fashion products with sustainability features? (1) Under 5%; (2) 5–10%; (3) 10–20%; (4) 20–50%; (5) Above 50%.

Specifically, the distribution of sustainable purchases of fast fashion products (Q12) is quite similar for both males and females (see Figure 3) with males a bit more skewed to the left. This is consistent with the findings in cognitive questions, where males also have a bit higher awareness of sustainability issues.

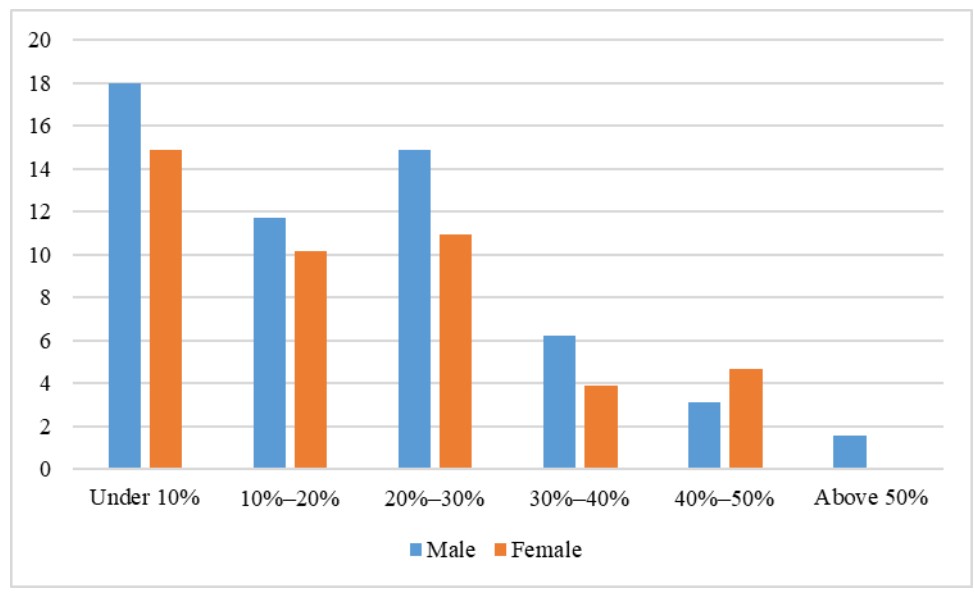

**Figure 3.** Distribution of Sustainable Purchases by Gender.

In contrast, the price sensitivity of sustainable fast fashion products is significantly different across gender. Most males are willing to pay 5–10% more to buy fast fashion brands with sustainable features (Figure 4), while females are likely to pay a higher premium for their purchases (10–20%). In other words, females are less sensitive to prices when purchasing fast fashion products with sustainability features.

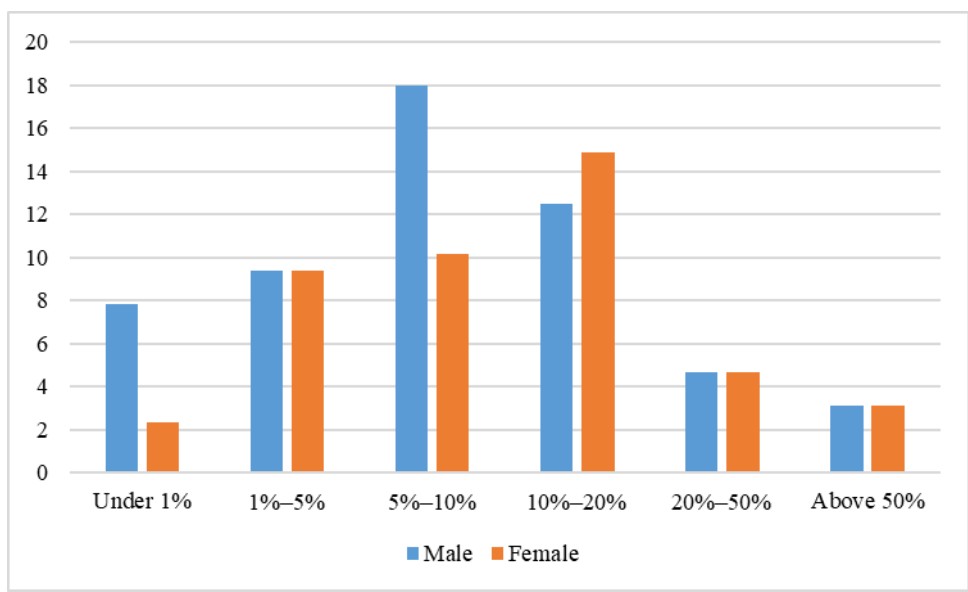

**Figure 4.** Distribution of Price Sensitivity of Sustainable Purchases by Gender.

By contrast, the income sensitivity follows an opposite pattern to the price sensitivity (Figure 5). Males are more ready to increase their purchases of sustainable fast fashion products than females. This finding is again consistent with the findings in sustainability awareness.

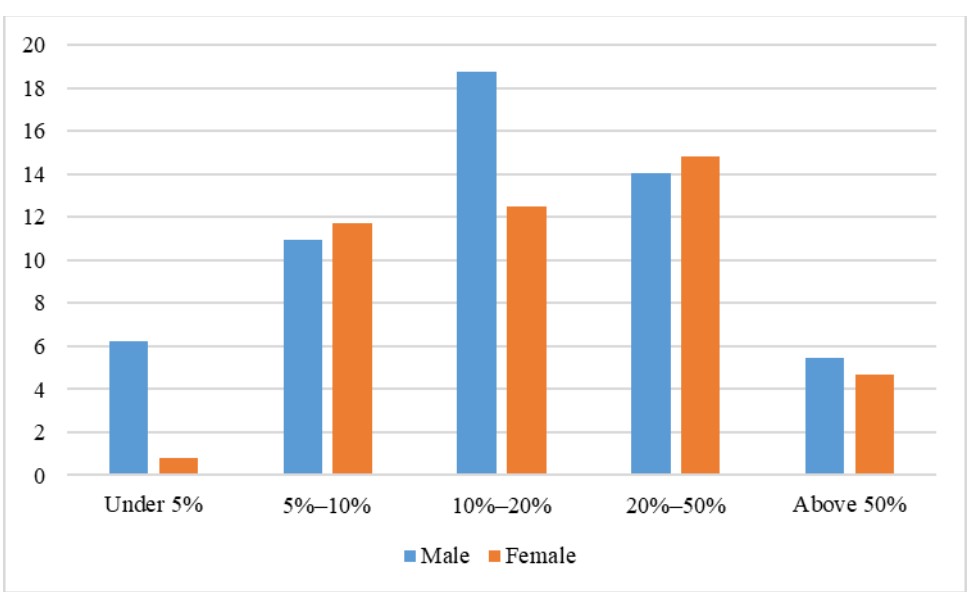

**Figure 5.** Distribution of Income Sensitivity of Sustainable Purchases by Gender.

In addition to the behavioural questions, Q15 and Q16 are asked to study the effects of friends and social media on fast fashion consumption behaviour.

- Q15: My friends and/or family affect my purchase of fashion products with sustainability features.
- Q16: Social media influencers affect my purchase of fashion products with sustainability features.

It is found that females (3.398) are more likely to be affected by friends/family and social media than males (3.155), and social media are more influential than friends/family, but the mean is quite low (the scale ranges from 1 to 5, with 3 being neutral).

The analyses above are descriptive and intuitive, but what is omitted from the analysis is the determinants for these behavioural indicators. To quantify contributions of the determinants, we use a similar oprobit model from the last subsection, but augmented with two indices to control for the cognitive factor (C = a simple average of cognitive questions, Q7–Q10) and the affective factor (A = a simple average of affective questions, Q17–Q20). The results are reported in Table 5.

**Table 5.** Oprobit Regression Results for Behavioural Questions.

|  | **Regressors** | **Q12** | **Q13** | **Q14** |
|---|---|---|---|---|
| Q1 | Female | −0.46 ** | 0.33 ** | 0.19 ** |
| Q2 | Age | 0.03 | −0.099 * | 0.00 |
| Q3 | European | 0.04 | 0.53 | 0.09 |
|  | American | 0.873 * | 0.37 | 0.10 |
|  | Oceanian | −0.10 | 0.839 * | −0.44 |
|  | African | −0.02 | −0.30 | 0.00 |
|  | Asian | −0.299 | 0.271 | 0.664 * |
| Q4 | Islam | −0.352 | −0.086 | 0.123 |
|  | Buddhism | −5.704 ** | −0.103 | 1.084 * |
|  | Hindu | 1.816 * | 0.474 | −1.599 * |
|  | Sikhism | −0.199 | 0.091 | −0.145 |
|  | No Religion | 0.009 | −0.175 | 0.195 |
| Q5 | Part-time worker | −0.057 | 1.032 * | −0.366 |
|  | Self-employed | 0.215 | 6.942 ** | −6.514 ** |
|  | Unemployed | −0.774 * | −1.232 * | −0.333 |
|  | Student | 0.281 | 0.506 | −0.56 |
|  | Other | 0.079 | 0.697 | −0.026 |
| Q6 | Budget Share | −0.093 | −0.375 | −0.675 |
| Q15 | Friends | 0.375 | 0.023 | 0.012 |
| Q16 | Social Media | 0.482 * | 0.033 | 0.013 |
| C | Cognitive Index | 0.087 ** | −0.019 * | −0.135 ** |
| A | Affective Index | 0.244 *** | 0.016 * | −0.187 ** |
|  | No. of Obs. | 128 | 128 | 128 |
|  | AIC | 417.815 | 442.384 | 399.056 |
|  | BIC | 486.263 | 510.833 | 464.653 |

Significance levels: * 10%, ** 5%, *** 1%.

To summarise, we list the key findings on the behavioural component of the attitude towards sustainability (Research Objective 2) here:

- Females are more susceptible to friends/family and social media than males in consumption of fast fashion products with sustainability features.
- Those who work behave significantly differently from those who do not.
- Social media have greater influence on consumption behaviour than word-of-mouth from friends and family.
- Compared to males, females have a lower price sensitivity and a higher income sensitivity.
- The cognitive index and the affective index play significant roles in determining the consumption decisions for fast fashion products.

### 4.3. Affective

Economics treat consumers as rational, but people also have feelings. This subsection focuses on the affective dimension of fast fashion purchases related to sustainability (Research Objective 3).

Four affective questions are asked to study their feelings regarding sustainability in the fast fashion industry:

- Q17: I feel disgusted when I learn how much waste and pollution are generated by fast fashion industry.
- Q18: I feel angry when I learn about labour slavery and child labour in fast fashion global supply chain.
- Q19: I feel honoured if I choose a fashion brand that engages in promoting sustainability.
- Q20: I feel interested in a fashion brand that engages in promoting sustainability.

Figure 6 compares the mean of the answers to affective questions by gender. The answers are ordered long a 5-sale metric: (1) strongly disagree, (2) disagree, (3) neutral, (4) agree, and (5) strongly agree. The overall attitude towards sustainability is positive (greater than 3). Nevertheless, it is shown that the difference is insignificant cross genders and across issues.

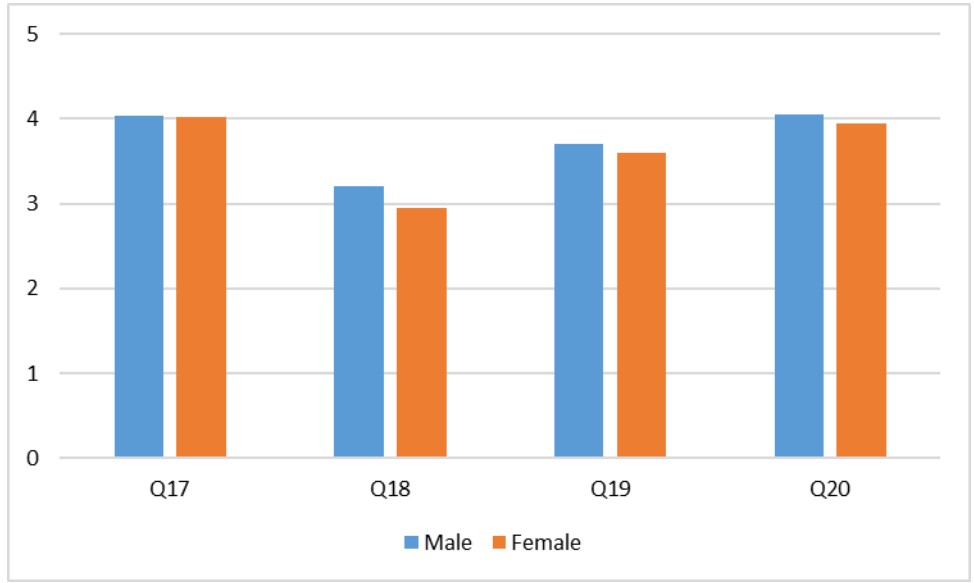

**Figure 6.** Distribution of Affective Questions by Gender.

Again, the information above is purely descriptive. To investigate the effects of different factors on the affective component, we employ the oprobit model to model the ordinal dependent variables. The estimation results are reported in Table 6. The key findings on Research Objective 3 are summarised here:

- Males and females share a very similar pattern in affective component of attitude towards sustainability issues.
- Environmental issues receive greater concerns compared to labour slavery and child labour in the fast fashion industry.
- Cultural and religious background plays an essential role in determining the feeling of fast fashion products and brands which engage in promoting sustainability.
- Employment status contributes to how customers feel about sustainability in purchasing fast fashion products.
- The cognitive index is positively related to the affective attitude towards sustainability in fast fashion purchases.

**Table 6.** Oprobit Regression Results for Affective Questions.

|  | **Regressors** | **Q17** | **Q18** | **Q19** | **Q20** |
|---|---|---|---|---|---|
| Q1 | Female | −0.119 * | −0.24 | 0.061 | −0.103 |
| Q2 | Age | 0.044 | −0.075 * | −0.077 | −0.021 |
| | European | 0.056 ** | 0.051 ** | −0.1 ** | −0.209 ** |
| | American | 0.038 | 0.045 ** | −0.893 ** | −0.794 ** |
| Q3 | Oceanian | −0.602 | −0.112 | −0.291 | 0.094 |
| | African | −1.007 *** | 0.055 ** | −0.011 | 0.092 |
| | Asian | 0.106 | 0.28 | 0.078 | 0.059 |
| | Islam | −0.502 ** | −0.27 ** | 0.107 ** | −0.139 |
| | Buddhism | −0.743 ** | 0.279 ** | −1.339 ** | −0.001 |
| Q4 | Hindu | −0.441 | 0.81 ** | −0.017 | −0.064 |
| | Sikhism | −0.11 | −0.556 ** | −0.187 | −0.007 |
| | No Religion | −0.927 ** | −0.697 ** | 0.731 ** | −0.58 ** |
| | Part-time worker | −0.378 * | 0.77 ** | 0.826 | 0.546 |
| | Self-employed | 2.154 *** | 5.014 *** | −1.114 *** | 5.214 *** |
| Q5 | Unemployed | −1.694 *** | −0.214 | 0.733 ** | 0.569 |
| | Student | −0.655 | −0.819 ** | 0.889 ** | 0.205 |
| | Other | −0.293 | −0.771 ** | −0.284 | 0.386 |
| Q6 | Budget Share | 0.048 | 0.016 | −0.073 | −0.075 |
| | Cognitive Index | 0.898 ** | 0.279 ** | 0.285 * | 0.281 * |
| | No. of Obs. | 128 | 128 | 128 | 128 |
| | AIC | 324.004 | 429.874 | 390.228 | 325.788 |
| | BIC | 389.601 | 495.471 | 455.825 | 391.385 |

Significance levels: * 10%, ** 5%, *** 1%.

## 5. Discussion

This section connects the empirical findings on the three components of attitudes towards sustainability with the existing literature.

There are three important findings on the cognitive component of attitude towards sustainability. First, the females are less aware of the social equality issue, confirming the simple correlation analysis, but not significantly different from the male counterpart on child labour and environmental issues [25]. Second, those who work (part-time, full-time or self-employed) tend to be more aware of the sustainability issues than those who do not (unemployed or students) as found in [45]. Third, awareness drops as the budget share and knowledge of fast fashion products rises, supporting the consumer theory in economics and information search theory in marketing literature [65]. Other factors, such as age, nationality (cultural background) and religion, are not systematically significant.

Now turn to the behavioural component of attitude towards sustainability. The first finding is on the gender dimension. Females spend less on fast fashion products with sustainability features, but they are more stable in terms of the willingness to pay a higher price and the responsiveness to a pay rise [14]. Second, those who work tend to behave differently in purchasing fast fashion products from those who do not. Arguably, people with a higher income care about sustainability more than those with no jobs [75]. Third, both cognitive index and affective index contribute to the consumption decisions. This confirms some factors identified in economic, psychological, anthropological and marketing theories in the literature. This is a new finding added in the literature on sustainability. However, age, nationality and religion are still not significant. It is also surprising that the current generation are not affected by friends or social media as found in other empirical literature [66].

Surprisingly, the findings on the affective component of attitude towards sustainability are very different from the cognitive and behavioural questions. First, gender no longer matters as shown in the insignificant coefficients, which agrees with the descriptive results. In contrast, gender is a very important factor driving the difference in cognitive and

behavioural questions. Second, cultural background and religious background play a significant role in explaining the difference in people's feeling about sustainability in fast fashion products [75]. For example, Europeans are more concerned with environmental and slavery issues compared to others, while Americans do not give credit to fast fashion brands with sustainability features. African respondents care about slavery more than any other country. Muslims have significantly lower consideration of sustainability when they purchase fast fashion products. Buddhists are more concerned about social equity (labour slavery and child labour) than environmental issues. Hindus and Sikhs have opposite views on slavery. Third, employment status continues to be an important factor, and, again, those with higher income flow (self-employed) tend to value sustainability more [45]. They are also more interested in fast fashion brands with a sustainability element. In other words, they are more likely to be the target customers. Fourth and most importantly, the cognitive index (defined as the simple average of cognitive questions Q7–Q9) significantly contributes to the affective questions. If the respondent is more aware and knowledgeable of sustainability, then they tend to have a stronger feeling about products with anti-sustainability features. They are also more likely to give moral credit and show interest in fast fashion brands that engage in promoting sustainability. Therefore, the link between cognitive and affective mechanisms is verified [70].

## 6. Conclusions

To achieve the research aim (to explore consumer attitude towards sustainability of fast fashion products in the UK), a comprehensive theoretical literature in economics, psychology, anthropology and marketing is reviewed to establish a conceptual framework of attitude and its causing factors, based on which an online questionnaire is designed to collect data. Both descriptive statistics and regression analysis are employed to address the three research objectives, i.e., the affective component, behaviour component and cognitive component (or "ABC") of the consumer attitude. We summarise some key theoretical implications here.

It is observed that age is never a significant factor in determining any of the three components, which suggests that our sample is a homogenous set across the millennial generation (or the Gen Z). In contrast, employment status can explain a substantial part of the difference in attitude towards sustainability. Moreover, we see that cognitive and behavioural components tend to converge across cultures and religions, but the affective component is still disparate. Different issues receive different degrees of attention, but there is a rising trend for the young generation to put sustainability as an important role when purchasing fast fashion products. Last but not the least, affectional factors are more important than cognitive factors in determining purchase behaviour of fast fashion product with sustainability features.

Based on the theoretical implications, we can draw some managerial implications for consumers, producers, and policymakers of the fast fashion industry. First, for consumers of fast fashion products, we find that the claimed knowledge on sustainability is likely to be higher than the actual knowledge due to over-confidence, so the attitude towards sustainability can be overestimated. It is advisable for consumers to update their knowledge on a regular and continual basis to avoid being tricked by fake advertisements. Second, for producers of fast fashion products, they should be aware that customers, especially the female working class in the UK, are not sensitive to fast fashion products with sustainability features. Therefore, excessive emphasis on that would probably do harm to the market share. Given that the female, young working class constitute a major part of the fast fashion market, it is a very important implication for marketing strategies of fashion producers. Finally, despite good public awareness of sustainability, policymakers should see that awareness does not automatically translate into economic behaviour. It is therefore vital for government policies to set proper taxes and subsidies to encourage a more sustainable fast fashion industry in the long run.

This paper of course has some limitations for future studies to overcome. The first is the data coverage. Due to the constraints of resources and the COVID-19 pandemic, questionnaires cannot be collected face-to-face. The response rate is usually higher if it can be done in person. The sample size in our study is not big, but big enough to draw statistically meaningful conclusions. Second, the questionnaire is only sent to university students and graduates in Wales, which may not reflect the young generation's attitude. Nevertheless, the young university students and professionals are arguably the main fast fashion products buyers [7]. According to the ONS [74], only 42% of the population in the UK has an undergraduate degree, which means our sample may not represent those without degrees. Nevertheless, it is found in the literature that Gen Z have a very similar pattern in their attitude towards sustainability due to the permeation of the Internet and social media [76]. Therefore, we believe our findings are generalisable and reliable. Future studies can verify this conjecture with a wider sample covering non-degree holders. The third limit is the quantitative method employed in data analysis. Regressions can well isolate the effects of different factors, but the research is more confirmatory in nature. Alternative methods like in-depth interviews and experiments can be used to explore how attitude is formed and transmitted.

**Author Contributions:** Conceptualization, B.Z., Y.Z. and P.Z.; methodology, B.Z., Y.Z. and P.Z.; software, P.Z.; validation, B.Z., Y.Z. and P.Z.; formal analysis, B.Z., Y.Z. and P.Z.; investigation, B.Z., Y.Z. and P.Z.; resources, B.Z.; data curation, B.Z., Y.Z. and P.Z.; writing—original draft preparation, B.Z., Y.Z. and P.Z.; writing—review and editing, P.Z.; visualization, B.Z., Y.Z. and P.Z.; supervision, P.Z.; project administration, B.Z.; funding acquisition, B.Z. All authors have read and agreed to the published version of the manuscript.

**Funding:** This research was funded by National Natural Science Foundation of China, grant number 71702116; Fundamental Research Funds for the Central Universities, grant number buctrc202022; and Key project of Beijing Municipal Education Commission, grant number SZ20171003824.

**Institutional Review Board Statement:** The study was conducted according to the guidelines of the Declaration of Helsinki and approved by the Ethics Committee of Cardiff Metropolitan University (date of approval 15 September 2020).

**Informed Consent Statement:** Informed consent was obtained from all subjects involved in the study before the questionnaire was filled online.

**Data Availability Statement:** The data are collected using online questionnaire (Google Form) and are anonymously collected after ethical approval. The regression codes and results are available on request.

**Acknowledgments:** We are grateful for the useful comments provided by Jo Tidy and ethical approval by Oeppen Hill of Cardiff Metropolitan University.

**Conflicts of Interest:** The authors declare no conflict of interest. The funders had no role in the design of the study; in the collection, analyses, or interpretation of data; in the writing of the manuscript, or in the decision to publish the results.

## Appendix A

The online questionnaire using Google Form is attached below.
Consumer Attitude towards Sustainability of Fast Fashion Products in the UK.

*If you are happy to participate in this project, please complete the following questionnaire. Participation is voluntary and participants are free to withdraw at any time. All data will be kept confidential and anonymous. This questionnaire has 20 questions and will take no more than 5 min to complete.*

*No prior knowledge of sustainability is required in order to participate but you need to be over 18 years old and have purchased fast fashion products such as Zara, H&M, Primark, Topshop etc.*

*Section 1: Individual Attributes.*

1. *What is your gender?*

- *Male*
- *Female*
- *Other*

2. *Which year were you born? (e.g., 1995) ____*
3. *What is your nationality?*

- *British*
- *Other European*
- *American*
- *Oceanian*
- *African*
- *Asian*

4. *What is your religion?*

- *Christian*
- *Muslin*
- *Buddhism*
- *Hindu*
- *Sikhism*
- *No religion*
- *Other*

5. *What is your current employment status?*

- *Full-time employed*
- *Part-time employed*
- *Self employed*
- *Unemployed*
- *Home maker*
- *Student*
- *Retired*
- *Other*

6. *What is your budget share in fast fashion products?*

- *Under 10%*
- *10–20%*
- *20–30%*
- *Above 30%*

### Section 2: Sustainability Knowledge.

7. *I am aware of social equity issues in the fast fashion industry such as working conditions of factory worker and fair trade. Strongly Agree, Agree, Neutral, Disagree, Strongly Disagree*
8. *I am aware of child labour and sweatshop issues in the global supply chain of the fast fashion industry. Strongly Agree, Agree, Neutral, Disagree, Strongly Disagree*
9. *I am informed of environmental issues in the fast fashion industry such as waste and pollution caused by excessive production of garments. Strongly Agree, Agree, Neutral, Disagree, Strongly Disagree*
10. *I am knowledgeable about the apparel brands that sell eco-friendly fashion products. Strongly Agree, Agree, Neutral, Disagree, Strongly Disagree*
11. *To your knowledge, which of the following fabrics is the most eco-friendly?*

- *Cotton*
- *Synthetics*
- *Wool*
- *Modal*

### Section 3: Sustainability Decision.

12. *From all your clothing purchases, what percentage is sustainable?*

- *Under 10%*

- *10–20%*
- *20–30%*
- *30–40%*
- *40–50%*
- *Above 50%*

13. *How much percent more would you like to pay for a fast fashion product with sustainability features?*
    - *Under 1%*
    - *1–5%*
    - *5–10%*
    - *10–20%*
    - *20–50%*
    - *Above 50%*

14. *How much percent more income is needed before you can consider fashion products with sustainability features?*
    - *Under 5%*
    - *5–10%*
    - *10–20%*
    - *20–50%*
    - *Above 50%*

15. *My friends and/or family affect my purchase of fashion products with sustainability features? Strongly Agree, Agree, Neutral, Disagree, Strongly Disagree*

16. *Social media influencers affect my purchase of fashion products with sustainability features? Strongly Agree, Agree, Neutral, Disagree, Strongly Disagree*

**Section 4: Sustainability Feeling.**

17. *I feel disgusted when I learn how much waste and pollution are generated by fast fashion industry. Strongly Agree, Agree, Neutral, Disagree, Strongly Disagree*

18. *I feel angry when I learn about labour slavery and child labour in fast fashion global supply chain. Strongly Agree, Agree, Neutral, Disagree, Strongly Disagree*

19. *I feel honoured if I choose a fashion brand that engages in promoting sustainability. Strongly Agree, Agree, Neutral, Disagree, Strongly Disagree*

20. *I feel interested in a fashion brand that engages in promoting sustainability. Strongly Agree, Agree, Neutral, Disagree, Strongly Disagree*

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
