# Peer review of "Consumer Attitude towards Sustainability of Fast Fashion Products in the UK"

_sustainability, doi:10.3390/su13041646_

Round 1
Reviewer 1 Report
The authors included my corrections in the new version.
I have no more comments.
Author Response
Thanks for approving our paper and effort.
Reviewer 2 Report
As stated in the last review, the paper needs some more improvements. The authors made some changes, but they are not enough.
The introduction fails to highlight properly the research gap; the research question; the theory where the paper adds value to existing literature; the connection between the research question and the theory; the connection between the research question and the research methodology; the novelty of the research; the last paragraph of the introduction should briefly introduce the next sections of the paper.
Section 2.3. is too extensive detailed compared to the other sections. In my opinion this section should be shorten and should be more comprehensive. It would also be good to cite some more up to date papers that deal with fashion and sustainability, for instance:
https://www.tandfonline.com/doi/abs/10.1080/17543266.2019.1572789
https://www.mdpi.com/2075-471X/8/4/24
https://www.mdpi.com/2071-1050/11/17/4532
https://www.mdpi.com/2071-1050/12/6/2477
Section 2.4. should be briefly included in the introduction and explained in more detail in the research methodology section
"The logic of our paper is as follows. First, the research context in the Introduction 376 section identifies the UK as a good case study for studying the customer attitude towards 377 sustainability in the fast fashion industry. Second, in the Literature Review section, we 378 review the three strands of literature to summarise different components and factors of 379 attitude. Third, this section operationalises the conceptual framework into a data strategy. 380 A carefully designed online questionnaire quantifies measures of the components and fac-381 tors of attitude towards sustainability. The next section therefore analyses the data col-382 lected and draw empirical conclusions." this section should be in the introduction. Why do you have it here at the methodology? What is the purpose of stating here what you have already done?
Please also present the operationalisation of your questionnaire!
The results are a mess. They are not logically presented at all. Results should not be presented depending on the questions from your questionnaire.
Discussions are now better
Limitations should be included in the conclusions section
Conclusions should consist of 4 paragraphs:
- theoretical implications
- managerial implications
- limitations of own research
- future research perspectives
Your present conclusions are not conclusions!
Reviewer 3 Report
Dear authors,
Thank you very much for your efforts writing this paper.
You have a depth understanding and broad background knowledge related to the topic of research. Nevertheless in my opinion this research is not clearly designed and it has not enough scientific quality to be published in a Q2 JCR journal.
This manuscript studies the consumer attitude towards sustainability of fast fashion products in the UK, which is an interesting and relevant topic of research.
The analysis is based on both descriptive statistics and ordered probit regression analysis.
The main contribution of this paper is to establish a tri-component model of attitude based on affective, behavioural and cognitive variables.
Results show that although there is an improved cognitive and affective awareness of sustainability, UK students and alumni do not automatically translate it to purchase behaviour.
The most relevant weaknesses of this investigation are, in the words of the authors:
- Line 726: “This paper limitation… is the data coverage”.
- Line 731: “second limit is the quantitative method employed in data analysis. Very simple oprobit regressions are used, without more sophisticated robustness check”.
- Line 734: “Alternative methods like in-depth interviews can also be employed to complement the regression analysis”.
Following comments aim to improve the quality of this paper. I primarily recommend addressing:
C1. Line 16: Please take into consideration that “alumni” is the plural of “alumnus”.
C2. Line 16: A sample of 128 responses is not enough representative. Additionally data only cover university students and alumni. The current population of the United Kingdom is 68,082,466 (UN, 2021) and most of them are neither students nor alumni (ONS, 2017).
C3. Line 91: “research aim and objectives”. Please clarify why you use both. In case it is uselessly repeated please eliminate one of them.
C4. Figures 1, 2 and 3: Usually figures are used to make easier to understand the results of our own research, not to show the contributions of other publications.
C5. Line 259: The Figure 2 “Maslow’s Hierarchy of Needs” is so popular that it is unnecessary to be shown. As it is in mind of every reader it would be enough to cite it.
C6. Line 350: Please improve your paper structure: don’t include subsection 2.4 Research questions in section 2 Literature review.
C7. Line 350: In the research questions section it is expected to find the research questions. Please take into consideration that a research question is a question. You can learn how to develop strong research questions here: https://www.scribbr.com/research-process/research-questions/
C8. Line 362: Please improve your paper structure: don’t include the research aim identification in the subsection of research questions.
Taking into consideration your high knowledge on this area, and your mathematical skills, I am sure you can write a new and much better paper.
Yours sincerely.
Round 2
Reviewer 2 Report
I think that most of the suggestions were implemented.
Reviewer 3 Report
Dear authors,
I would like to thank you for the significant improvement of your manuscript related to the consumer attitude towards sustainability of fast fashion products in the UK.
You took into consideration all my suggestions and clearly improved the manuscript: all issues have been resolved.
I congratulate the authors for this interesting investigation and wish them the most success in their research activities.
Thank you very much for your efforts and for your valuable scientific contribution.
Yours sincerely.
This manuscript is a resubmission of an earlier submission. The following is a list of the peer review reports and author responses from that submission.
Round 1
Reviewer 1 Report
The topic is interesting and up to date. It is relevant to the theory and practice. However, some improvements are needed to make the manuscript in line with journal publication standards.
1. Abstract: Please rewrite the abstract to include the most important information in it: reasons for choosing the topic, relevance to knowledge, contribution, basic assumptions and the most important results. The Abstract is too short and does not reflect well the contents of the paper. This section as well as Introduction should better emphasize the new contributions of the study.
2. From line 88, add a "literature review" section.
Please refer to the latest publications from "Sustainability":
1. Lysenko-Ryba, K.; Zimon, D. Customer Behavioral Reactions to Negative Experiences during the Product Return. Sustainability 2021, 13, 448.
2. Wei, A.-P.; Peng, C.-L.; Huang, H.-C.; Yeh, S.-P. Effects of Corporate Social Responsibility on Firm Performance: Does Customer Satisfaction Matter? Sustainability 2020, 12, 7545.
3. Zimon, D., Madzík, P., & Sroufe, R. (2020). The Influence of ISO 9001 & ISO 14001 on Sustainable Supply Chain Management in the Textile Industry. Sustainability, 12(10), 4282.
4. Zhang, R.; Li, J.; Huang, Z.; Liu, B. Return Strategies and Online Product Customization in a Dual-Channel Supply Chain. Sustainability 2019, 11, 3482
etc.
3. The theoretical positioning of this paper needs extension through arguing and developing hypotheses as part of the literature review.
4. How the sample was designed and approached to collect the data whether the sample is representative?
5. Results are narrowly presented and discussed. Some relations with the extant literature should be underlined and argued. Extract a discussion section and expand it (However, I will not insist on this. I leave it to the Authors)The linkage of the results with previous findings could improve the paper.
Good Luck!
Reviewer 2 Report
The paper is interesting and has some publication potential, however some adjustments and revisions must be done. The paper is not focused at all and it is really not clear what the authors want to achieve with this paper.
The entire article needs proofreading as some parts and phrases are not clear.
Abstract: "Paint" is not really the proper concept
The research question is not pointed out very good.
Please try to explain how the paper adds value to existing literature.
Introduction
"This figure is 16.7kg in Germany, 16.0kg in Denmark, 9.0kg " This is not the proper academic style.
this is more the research context "ashion industry is reportedly the world’s third biggest " which, I think, should be in the methodology as a subsection
"Among others, the “fast fashion” business model is a salient success in this trend, 37
because its low prices and fast product rotations encourage over-consumption. By the 38
mid-1970s, many fashion " ok, but from where do you know that? These ideas must be referenced.
"It is in the 1990s when the fast fashion industry became ma- 43
ture and many leading brands such as Zara, H&M and GAP have established " reference for that?
"opment Goals in September 2015. Three environmental initia- 66
tives are developed in the UK since then: waste disposal " reference for that?
" (Shen et al., 2014; Joy et 72
al., 2012)." when you have multiple references cite them chronologically.
The introduction fails to highlight properly the research gap; the research question; the theory where the paper adds value to existing literature; the connection between the research question and the theory; the connection between the research question and the research methodology; the novelty of the research; the last paragraph of the introduction should briefly introduce the next sections of the paper.
1.1. should be section 2. Literature review.
The literature on sustainability is not really the newest one. More references from 2018-2021 should be added.
Literature on consumer behaviour: I think that you should take into consideration the newest literature. Of course citing Marshall or Adam Smith or David Ricardo is nice, but I really doubt that they have anything do do with fast fashion. As this part is a literature review I would expect to have a more synthetised development of the consumer behaviour theory. It would be proper to cite some papers from the last years, especially those who took into consideration consumer behaviour in fashion, as fashion would be the next industry to fast fashion.
Smith and Ricardo are classical thinkers in economics. This paper is on business. Please try to discuss possible theories within business.
L 333-362: these aspects are not argued enough. They should be briefly mentioned in the introduction and explained at the methodology.
Methodology
For positive onthology... cite some references
This section should contain a better and more synthetised approach on the relevant literature. Please try to stress out why the research context of UK is so relevant, how data was collected, how the instrument was operationalised, how you have ensured that data is reliable. valid, and trustworthy.
Please also explain the logic of analysis that you conduct.
It would be good to stress out the conceptual framework of your paper.
Why have you considered young consumers? Which young consumers? Millennials? Gen Z?
Results:
it is not clear how results are presented.
Table 2: What scales are here? This table is not understandable. What have you performed here?
Usually a paper should have ONE research question and one or several research objectives. 7 research question for a paper... are TOOOOO MANY.
The results are a mess. They are not logcally presented at all.
The paper has no discussions. Comparisons with previous findings are missing.
Conclusions are very weak and almost not present.
The questionnaire is very weak. A proper analysis with this questionnaire is hard to obtain.
References: some of them are not properly displayed. Relevant references are missing. More newer references should be added.
Reviewer 3 Report
The topic and approach adopted are of interest and potential value in advancing our understanding and knowledge on a relevant and current topic
The article is well thought out and developed, furthermore, it is fully consistent with the aims of the journal.
Only a suggestion could be useful to improve the quality of this manuscript: I suggest to divide the discussion from the conclusion in order to dedicate more emphasis to concluding remarks.
Finally, in the end section, the authors should emphasize the main implications of their study.